# Domain generation algorithms detection with feature extraction and Domain Center construction

**Xinjie Sun**[1,2]*, **Zhifang Liu**[1]

1 Institute of Computer Science, Liupanshui Normal University, Liupanshui, Guizhou, China, 2 Guizhou Xinjie Qianxun Software Service Co., Ltd, Liupanshui, Guizhou, China

* sxj123lps@163.com

## Abstract

Network attacks using Command and Control (C&C) servers have increased significantly. To hide their C&C servers, attackers often use Domain Generation Algorithms (DGA), which automatically generate domain names for C&C servers. Researchers have constructed many unique feature sets and detected DGA domains through machine learning or deep learning models. However, due to the limited features contained in the domain name, the DGA detection results are limited. In order to overcome this problem, the domain name features, the Whois features and the N-gram features are extracted for DGA detection. To obtain the N-gram features, the domain name whitelist and blacklist substring feature sets are constructed. In addition, a deep learning model based on BiLSTM, Attention and CNN is constructed. Additionally, the Domain Center is constructed for fast classification of domain names. Multiple comparative experiment results prove that the proposed model not only gets the best Accuracy, Precision, Recall and F1, but also greatly reduces the detection time.

## 1 Introduction

Malware has now developed into the number one public enemy threatening network security. In order to avoid the detection of security facilities, its production process is becoming more and more complex. One typical approach is to integrate Domain Generation Algorithm (DGA) [1] into the software to generate a large number of rapidly changing domain names. As a backup or main means of communication with Command and Control (C&C) server, this method can effectively increase the robustness of botnet [2], so as to continuously control the infected host. Correspondingly, the research on DGA algorithm has becoming a hot topic of network security. However, due to the fast updating speed of DGA domains, existing research methods have too many false positives in practical use. Therefore, the detection of DGA domains is still an arduous task in the computer security field.

Discovering DGA domains is very important for maintaining network security. The existing solutions mainly include static blacklist [3], reverse engineering [4], machine learning [5] and deep learning [6]. Due to the slow update speed of static blacklist and the fast update speed of DGA domains, it is difficult to effectively apply static blacklist to DGA detection. Reverse

**Data Availability Statement:** All relevant data are within the paper and its Supporting information files.

**Funding:** Guizhou Province (Grant nos. KY[2020] 112), Liupanshui Normal University High level

Talent Research Launch Fund
[LPSSYKYJJ201708], Liupanshui Science and
Technology Bureau Fund Project[52020-2018-04-
15], Liupanshui Normal University Major
Comprehensive Reform Pilot Project
[LPSSYzyzhggsd202003], the Science and
Technology Foundation of Guizhou Province(ZK
[2022]528), the Youth Science and Technology
Talent Growth Project of Department of Education
in Guizhou Province (Grant nos. KY[2022]054).

**Competing interests:** The authors have declared
that no competing interests exist.

engineering requires a malware sample, which is not always feasible for DGA detection. In recent years, machine learning and deep learning methods provide new hope for DGA detection.

Machine learning and deep learning methods construct the feature set of domain names and combine machine learning or deep learning models to detect DGA domains. Since the deep learning model has stronger nonlinear modeling ability than the machine learning model, it can detect DGA domains more accurately. Although there have been some studies on detecting DGA domains through deep learning models [7–9], most of them only construct feature set through domain name. Due to the limited effective information contained in the domain name, the DGA domains can not be accurately detected. To overcome the above shortcomings, a feature set with rich features is constructed. Considering that the Whois [10] of the domain name contains rich features (e.g. registrar, registration time, etc.) related to the domain name category, the Whois features are extracted to construct the feature set. Considering that the N-gram [11] features of the domain name also contains rich information that can reflect whether the domain name is malicious, the blacklist and whitelist substring datasets of the domain name are constructed to obtain the N-gram features.

Due to the great ability in sequential modeling, Recurrent Neural Network (RNN) [12] is widely used in DGA detection among the vast type of deep learning models. Although RNN has made some achievements in DGA detection, it still has defects. Firstly, when the sequence is too long, the gradient disappearance problem inevitably occurs in RNN. Secondly, RNN assigns the same weight to all features. Finally, the high dimensionality of RNN makes the model difficult to converge. In the deep neural network, it has been proved that the more complex the neural network structure is, the better the effect is [13]. Therefore, Bi-Directional Long Short-Term Memory (BiLSTM), Attention and Convolutional Neural Network (CNN) are used to construct the DGA detection model. BiLSTM [14] is used to solve the gradient disappearance problem. Attention mechanism [15] is used to assign different weights to different features. CNN [16] is used to reduce the high dimensionality problem. In addition, skip connect [17] is used at the output of Attention network to solve the gradient disappearance and weight matrix degradation problems.

Although the classification results of the deep learning model will get better as the number of layers increases, the time spent will also increase [13]. Therefore, in order to reduce the DGA detection time on the validation set, the Domain Center is constructed. When there is a new domain name input, the feature vector of it is first obtained, then the hidden vector of it is obtained by the constructed deep learning model. Finally, the Euler distances [18] between the hidden vector and the mean vectors stored in the Domain Center are calculated to obtain the final classification results.

The main innovations of us are as follows:

1. A feature set including the domain name features, the Whois features and the N-gram features is constructed. To obtain the N-gram features, the domain name whitelist and blacklist substring feature sets are constructed.

2. A deep learning model based on BiLSTM, Attention, skip connect and CNN is constructed.

3. The Domain Center is proposed to reduce the DGA detection time on the validation set.

The remainder of this work is organized as follows. Section 2 introduces the latest research results of DGA detection; Section 3 introduces the background of BILSTM, Attention mechanism and CNN; Section 4 introduces the construction method of the feature set; Section 5 introduces the structure of the deep learning model constructed in this paper; Section 6 introduces the data set selected in this paper and the experimental results; Section 7 provides a final conclusion.

## 2 Related work

DGA detection methods include blacklist, reverse engineering, machine learning and deep learning. Although the blacklist method can provide effective security and is used by most network security companies [19, 20], its inherent defects in update speed make it easy for DGA domains to bypass the detection of blacklist. Reverse engineering requires a malware sample, which is not always feasible for DGA detection [21]. Therefore, most of the researches on DGA detection focus on machine learning and deep learning methods.

Machine learning method first construct DGA domain feature set, and then realizes DGA detection by machine learning models. Tuan et al. [22] proposed a machine learning based DGA detection model using TF-IDF and n-gram for feature representation. The results showed that logistic regression and SVM were the most effective. Štampar et al. [23] engineered a robust feature set, and accordingly trained and evaluated 14 ML, 9 DL, and 2 comparative models on two independent datasets. The experimental results showed that if ML features are properly engineered, there is a marginal difference in overall score between top ML and DL representatives. Soleymani et al. [24] applied machine learning algorithm and text mining technology to analyze DNS protocol and identify DGA botnets. The experimental results showed that the Random Forest could be effectively used in DGA botnet detection and had the best DGA botnet detection accuracy. Chin et al. [25] proposed a machine learning framework for identifying and clustering domain names to circumvent threats from the DGAs. Li et al. [26] proposed a machine learning framework (a two-level model and a prediction model) for identifying and detecting DGA domains to alleviate the threat of them. Baruch et al. [27] surveyed different machine learning methods for detecting DGAs by analyzing only the alphanumeric characteristics of the domain names in the network.

The deep learning method also needs to construct the feature set of DGA domains, and then realizes DGA detection through the deep learning models. Tuan et al. [28] proposed solutions for detecting and classifying DGA families. They proposed two deep learning models called LA_Bin07 and LA_Mul07 by combining the LSTM network and Attention layer. The experimental results showed that the LA_Bin07 and LA_Mul07 models solved the DGA botnets problem for binary and multiclass classification problems with very high accuracy. Namgung et al. [29] proposed an efficient DGA detection method based on BiLSTM, which further maximized the detection performance by using the CNN + BiLSTM integrated model, and allowed the model to learn local and global information at the same time in the domain sequence. The experimental results showed that the existing CNN and LSTM models had obtained F1 scores of 0.9384 and 0.9597 respectively, while the proposed BiLSTM and integrated model had obtained F1 scores of 0.9618 and 0.9666 respectively. Liang et al. [30] proposed three feature extraction methods adapted to the length of the DGA domains. In addition, they further analyzed the public suffix to evaluate its impact on the detection of DGA domains. The experimental results showed that the method greatly improved the detection performance. Lison et al. [31] demonstrated that a deep learning approach based on RNN was able to detect domain names generated by DGAs with high precision. Xu et al. [32] combined n-gram and a deep CNN to propose a novel n-gram combined character based domain classification (N-CBDC) model. Experiments on real-world data showed that N-CBDC could effectively detect DGAs. Ren et al. [33] proposed a deep learning framework for identifying and detecting DGA domains.

It can be seen from previous researches that the primary task of both machine and deep learning methods is to construct the DGA domain feature sets, which plays a key role in DGA detection. Considering that the existing researches basically construct feature set based on domain name only, which contains limited features, domain name features, Whois features and N-gram features are combined to construct feature set containing rich features.

## 3 Background

### 3.1 LSTM

LSTM [34] is a variation of RNN [12], and the structure of LSTM is shown in Fig 1. LSTM includes an input gate $i_t$, a forget gate $f_t$ and an output gate $o_t$. The forget gate $f_t$ accepts the output of the previous unit module $C_{t-1}$ and decides which part to keep and forget, which is calculated as follows [34]:

$$f_t = \sigma(W_f x_t + U_f h_{t-1} + b_f) \tag{1}$$

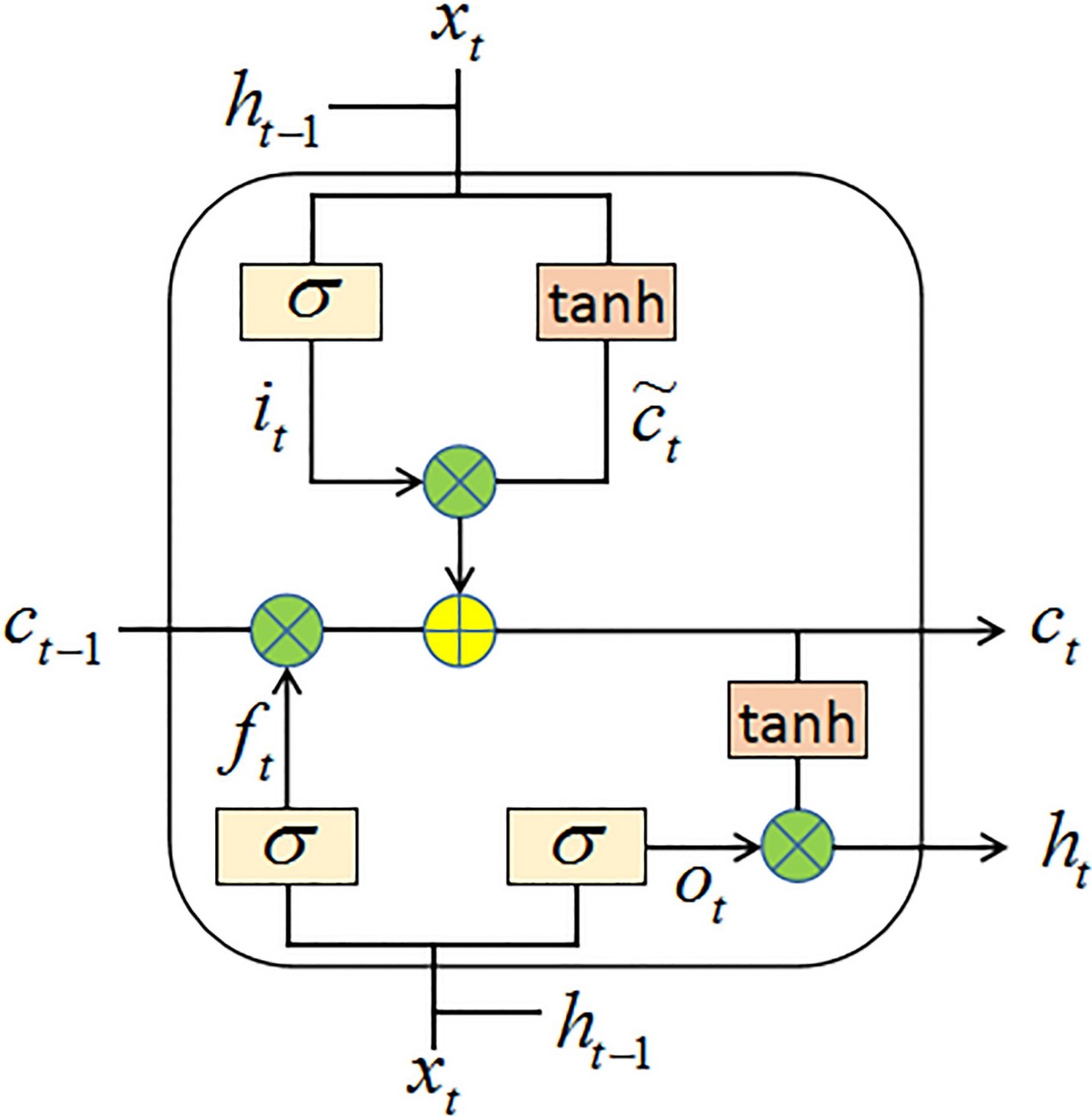

**Fig 1. The structure of LSTM.**

Where $x_t$ is the current input, $\sigma()$ is the element-wise sigmoid function, $W_f$ and $U_f$ are the weight matrices and $b_f$ is the bias term. The input gate $i_t$ determines which information is recorded into the cell state, and the cell state $c_t$ is obtained by merging $i_t$ and the new memory $\tilde{c}_t$. The formula of the input gate is as follows [34]:

$$i_t = \sigma(W_i x_t + U_i h_{t-1} + b_i) \tag{2}$$

$$\tilde{c}_t = tanh(W_c x_t + U_c h_{t-1} + b_c) \tag{3}$$

$$c_t = f_t \otimes c_{t-1} + i_t \otimes \tilde{c}_t \tag{4}$$

Where $W_i$, $W_c$, $U_i$ and $U_c$ are the weight matrices, $b_i$ and $b_c$ are the bias terms, $\otimes$ is the element-wise multiplication, $tanh()$ is the activation function Relu. The output gate $o_t$ determines the output value based on the cell state $c_t$. A Sigmiod function is first used to determine which part of $c_t$ needs to be output, then $c_t$ is processed through the $tanh()$ layer, and finally $o_t$ and $tanh(c_t)$ are multiplied to get the final desired output. Which can be denoted by [34]:

$$o_t = \sigma(W_o x_t + U_o h_{t-1} + b_o) \tag{5}$$

$$h_t = o_t \otimes tanh(c_t) \tag{6}$$

Where $W_o$ and $U_o$ are the weight matrices and $b_o$ is the bias term.

### 3.2 Attention mechanism

As the length of input sentences increases, the ability of LSTM to remember connections between words that are too far apart in a sentence decreases. Attention mechanism [15] solves the above problem by considering all input words to create a context vector and assigning relative weights to them. The structure of the Attention mechanism is shown in Fig 2. In Fig 2, $x = (x_1, x_2, \cdots, x_T)$ represents the input of LSTM, $h = (h_1, h_2, \cdots, h_T)$ represents the output through hidden layer of LSTM. The correlation $e_{tj}$ between the $jth$ input $h_j$ and the current hidden state $s_{t-1}$ is calculated as follows [15]:

$$e_{tj} = score(s_{t-1}, h_j) \tag{7}$$

Where $score()$ is a correlation operator, and the weighted dot product is chosen in this paper. A softmax transformation is performed on $e_{tj}$ to obtain the corresponding probability $a_{tj}$, which is calculated as follows [15]:

$$a_{tj} = \frac{exp(e_{tj})}{\sum_{k=1}^{T} exp(e_{tj})} \tag{8}$$

The context vector $c_i$ of time step $i$ is obtained by weighting the sum of $a_{tj}$ as follows [15]:

$$c_i = \sum_{j=1}^{T} a_{tj} h_j \tag{9}$$

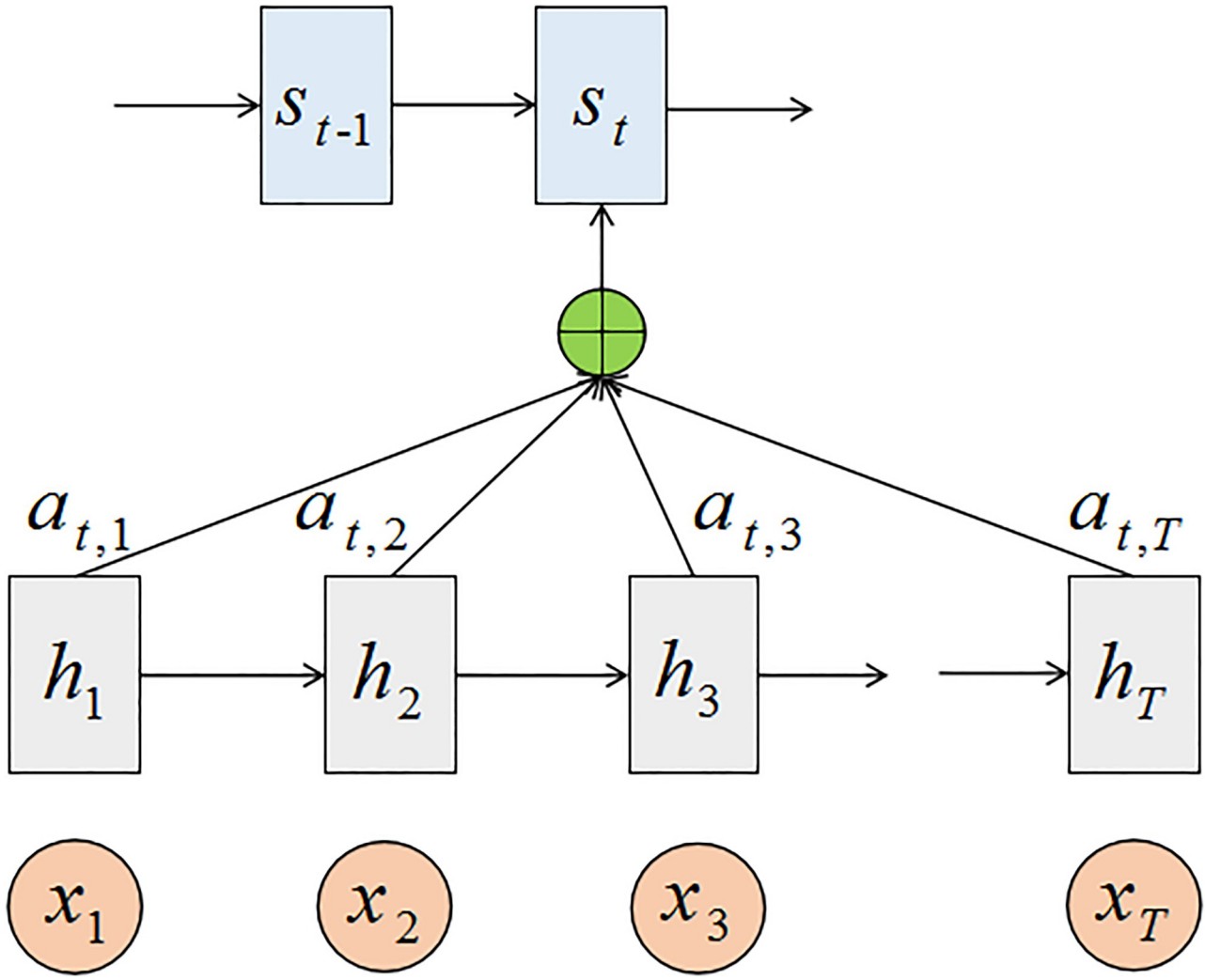

**Fig 2. The structure of Attention mechanism.**

### 3.3 CNN

CNNs [16] capture local information and reduce dimensionality by one-dimensional (1D) convolution and pooling operations. The structure of the CNN for text classification is shown in Fig 3. In the convolution layer, features are extracted with the help of various filters. Intermediate procedures are applied between the convolution layer and the pooling layer to make features nonlinear with the help of linear unit activation functions. In the pooling layer, these feature graphs are reduced in size, reducing the computational effort of subsequent layers and displaying important features more efficiently.

## 4 Construction of feature sets

### 4.1 Domain name feature set

As shown in Fig 4, a complete domain name includes the Top-Level Domain (TLD), the Second-Level Domain (SLD), and so on, which can reflect the category or attribution of the

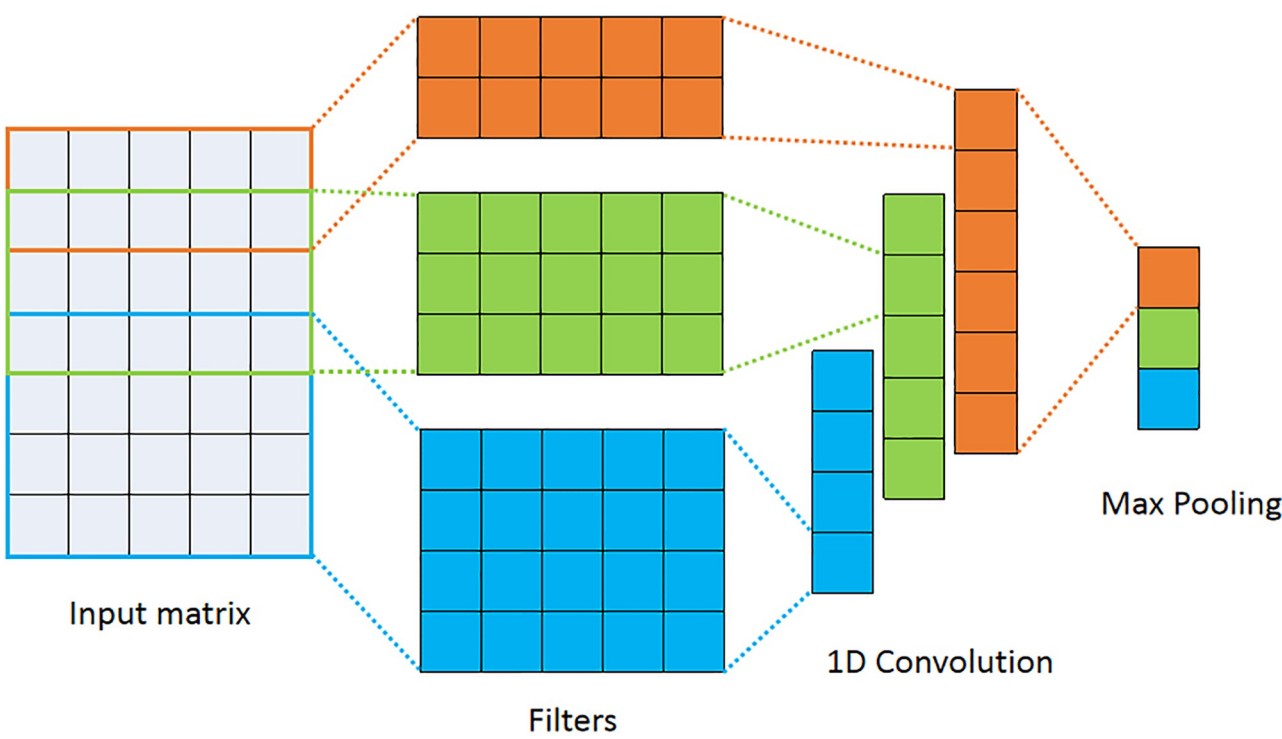

**Fig 3. The structure of CNN.**

website or the specific purpose of the domain name. Therefore, the domain name features shown in Table 1 are obtained.

## 4.2 Whois feature set

Whois domain name database stores the information of all registered domain names. Whois information can be used to check the availability of domain names, identify trademark infringement, and hold domain name registrants accountable. Rich information such as

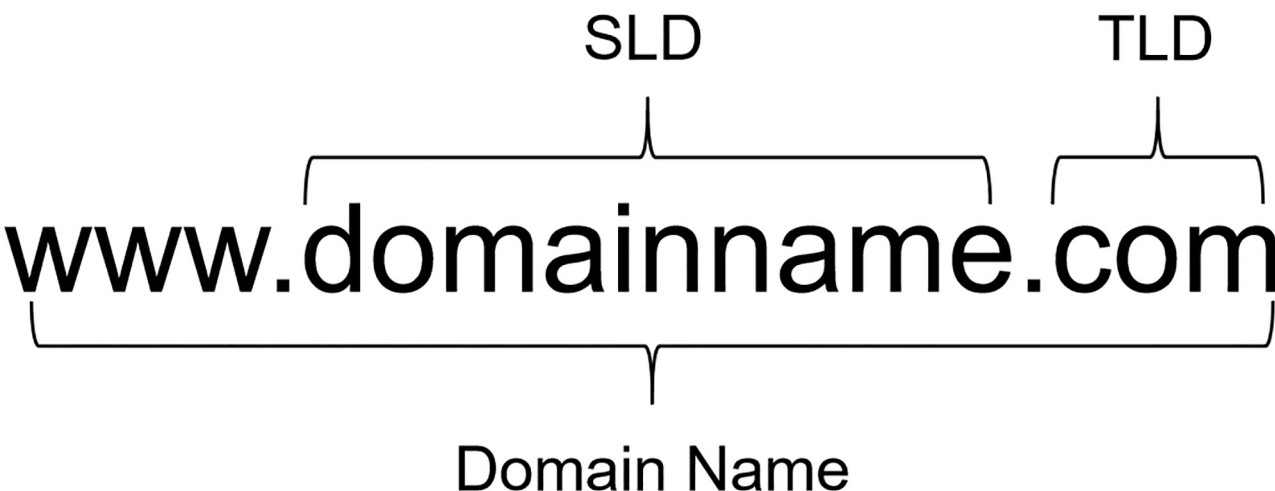

**Fig 4. The structure of domain name.**

**Table 1. Obtained domain name features.**

| FeatureName | Description | Type |
|---|---|---|
| Dom_1 | Category of TLD | int |
| Dom_2 | Length of SLD | int |
| Dom_3 | Layers of domain name | int |
| Dom_4 | Number of digits in domain name | int |
| Dom_5 | Proportion of digits in domain name | int |
| Dom_6 | Number of special symbols in domain name | int |
| Dom_7 | Proportion of special symbols in domain name | int |

registrant, registration time, DNS and so on can be obtained through Whois. As shown in Table 2, the Whois features are obtained and converted into int data.

### 4.3 N-gram feature set

**4.3.1 Domain name whitelist substring N-gram feature set.** Alexa ranking represents the world ranking of website popularity. Alex top 1 million stores the top 1 million websites in order of popularity, therefore, the top ranked websites in Alex top 1 million are usually the ones with higher credibility. Therefore, top 100,000 domain names of Alex top 1 million are selected to build the domain name whitelist substring feature set (DNWSFS), and the number of substrings of each domain name to be tested appared in the DNWSFS is obtained. The specific process is as follows.

Step one, remove the special characters of the 100,000 domain names of Alex top 1 million, and split them into substrings through the N-gram method. N-gram slides from the left to the right of the domain name using a sliding window of length N. Taking "domainname" as an example, the process of 4-gram is shown in Fig 5. According to the empirical value, the values of N are set to 3–8, and the substrings of 3-gram to 8-gram of the 100,000 domain names in Alex top 1 million are obtained respectively.

**Table 2. Obtained Whois features.**

| FeatureName | Description | Type |
|---|---|---|
| Who_1 | Is there a registrar | int |
| Who_2 | Is there a phone | int |
| Who_3 | Is there a mailbox | int |
| Who_4 | Is there an update time | int |
| Who_5 | Is there a creation time | int |
| Who_6 | Is there an expiration time | int |
| Who_7 | Is there a domain name server | int |
| Who_8 | Is there a domain name status | int |
| Who_9 | Country of Registrar | int |
| Who_10 | Telephone ownership | int |
| Who_11 | Mailbox category | int |
| Who_12 | Update time | int |
| Who_13 | Creation time | int |
| Who_14 | Expiration time | int |
| Who_15 | Type of domain name server | int |
| Who_16 | Domain name status | int |

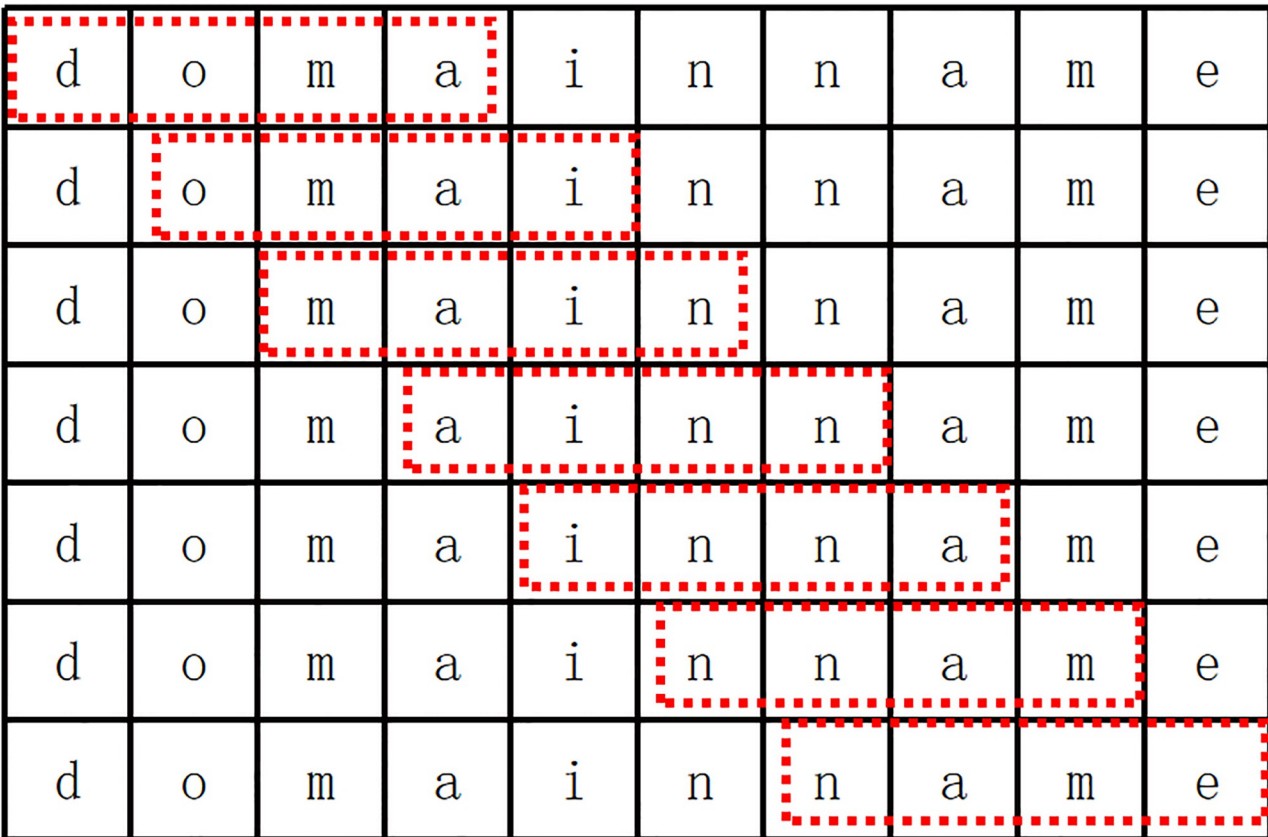

**Fig 5. Substring acquisition process of N-gram (N = 4).**

Step two, count the number of occurrences of each substring in step one, so as to build the DNWSFS. DNWSFS contains all the substrings and the number of times they occur.

Step three, remove the characteristic characters of each domain name to be tested, and obtain the substrings of 3-gram to 8-gram of them.

Step four, query the number of occurrences in DNWSFS of the 3-gram to 8-gram substrings of the domain name to be tested.

According to the above process, the domain name whitelist substring N-gram features are obtained as shown in Table 3.

**4.3.2 Domain name blacklist substring N-gram feature set.** The 360netlab dataset stores DGA domains screened in real time by researchers of 360 security company, containing domain names of 58 DGA families. In the same way as constructing the DNWSFS, 100,000 domain names of 360Netlab are selected to build the domain name blacklist substring feature set (DNBSFS), and the domain name blacklist substring N-gram features are obtained as shown in Table 4.

## 5 Method

The publicly available data are selected in this artical and all data can be publicly accessed by everyone. The domain names of Alex top can be downloaded from: https//www.alexa.com/. The DGA domain names can be downloaded from: https://data.netlab.360.com/.

**Table 3. Obtained domain name whitelist substring N-gram features.**

| FeatureName | Description | Type |
|---|---|---|
| Whi_1 | number of occurrences in DNWSFS of the 3-gram substrings of the domain name to be tested | int |
| Whi_2 | number of occurrences in DNWSFS of the 4-gram substrings of the domain name to be tested | int |
| Whi_3 | number of occurrences in DNWSFS of the 5-gram substrings of the domain name to be tested | int |
| Whi_4 | number of occurrences in DNWSFS of the 6-gram substrings of the domain name to be tested | int |
| Whi_5 | number of occurrences in DNWSFS of the 7-gram substrings of the domain name to be tested | int |
| Whi_6 | number of occurrences in DNWSFS of the 8-gram substrings of the domain name to be tested | int |

DGA detection model based on feature extraction and Domain Center construction, FEDCC, is constructed in this paper, and the structure of the model is shown in Fig 6. Firstly, as described in Section 4, the domain name features, Whois features and N-gram features of the input layer are extracted to form into a feature vector of domain name. Secondly, the feature vector is input into BiLSTM network to obtain the hidden vector. Thirdly, the Attention network is used to assign different degrees of attention to the hidden vector. Fourthly, the feature vector and the hidden vector output by the Attention network are added as the result of the skip connect network. Fifthly, the result of the skip connect network is input into the CNN network. 1D convolution is used to further extract hidden relationships, and max pooling is used to reduce dimension. Sixthly, the output of CNN network is input into the fully connected network, and the final classification result is obtained through the softmax function. Finally, The hidden vectors of all samples through the CNN network are input to the Domain Center, and the mean vectors of different categories of samples are further obtained. When a new domain name is input, it is only necessary to calculate the Euler distances between the hidden vector obtained by the deep learning model of the domain name and the mean vectors stored in the Domain Center to quickly achieve classification. The specific process is as follows.

## 5.1 BiLSTM network

Unlike LSTM [34], which can only use the information before the current time node, BiLSTM [14] can use the forward and backward timing characteristics at the same time. As described in Section 4, feature vector $V = \{v_1, v_2, \cdots, v_n\}$ with length $n$ is obtained. Then, $V$ is input into the forward and backward networks of BiLSTM to obtain the forward and backward hidden

**Table 4. Obtained domain name blacklist substring N-gram features.**

| FeatureName | Description | Type |
|---|---|---|
| Bla_1 | number of occurrences in DNBSFS of the 3-gram substrings of the domain name to be tested | int |
| Bla_2 | number of occurrences in DNBSFS of the 4-gram substrings of the domain name to be tested | int |
| Bla_3 | number of occurrences in DNBSFS of the 5-gram substrings of the domain name to be tested | int |
| Bla_4 | number of occurrences in DNBSFS of the 6-gram substrings of the domain name to be tested | int |
| Bla_5 | number of occurrences in DNBSFS of the 7-gram substrings of the domain name to be tested | int |
| Bla_6 | number of occurrences in DNBSFS of the 8-gram substrings of the domain name to be tested | int |

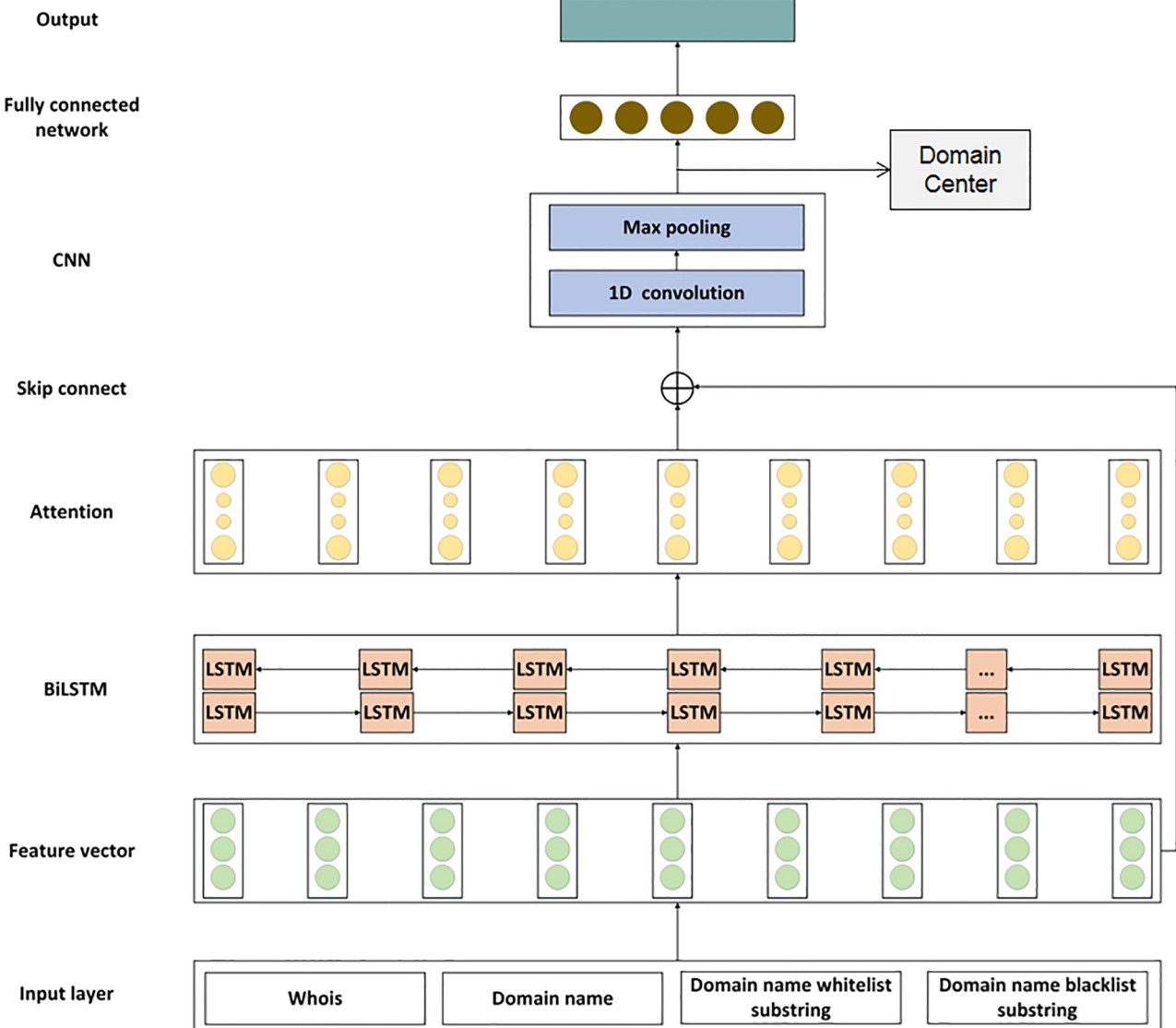

**Fig 6. The structure of the proposed model.**

vectors $\vec{h}, \overleftarrow{h}$:

$$\vec{h} = \overrightarrow{LSTM}(V) \qquad (10)$$

$$\overleftarrow{h} = \overleftarrow{LSTM}(V) \qquad (11)$$

Then, the forward and backward hidden vectors are combined to obtain the bidirectional hidden vector $h$:

$$h = [\vec{h}, \overleftarrow{h}] \qquad (12)$$

## 5.2 Attention network

Attention mechanism assigns different attentions to different features. Specifically, firstly, input $h$ into the Attention network to obtain the hidden representation $x$:

$$x = tanh(Wh + b) \tag{13}$$

Where $W$ represents the weight and $b$ represents the bias term. Secondly, the attentions of the features are calculated according to the similarity between $y$ and $x$, where $y$ is the randomly initialized feature vector. After the attentions are obtained, the softmax function is used for normalization, and then the weight vector $r$ is obtained:

$$r = \frac{exp(xy)}{\sum exp(xy)} \tag{14}$$

Finally, comment vector $f$ containing all feature concerns is obtained through the weighted sum of $r$:

$$f = \sum rh \tag{15}$$

## 5.3 Skip connect network

As the number of network layers deepens, the objective function is more and more likely to fall into local optimal solutions, while the problems of weight matrix degradation and gradient disappearance becomes more serious. Since the skip connect network [35] directly takes the input data as part of the output, it can alleviate the above problems well. Add $V$ and $f$ to obtain the result $V_{sk}$ of skip connect:

$$V_{sk} = V + f \tag{16}$$

## 5.4 CNN network

CNN network includes 1D convolution and max pooling. First, input $V_{sk}$ into the 1D convolution network to further extract hidden relationships in hidden vector:

$$t = relu(W_{cnn}f + b_{cnn}) \tag{17}$$

Where $W_{cnn}$ represents the weight and $b_{cnn}$ represents the bias term. Then, input $t$ into the max pooling network to obtain the pooling operation result $g$:

$$g = max\{t\} \tag{18}$$

## 5.5 Output

Input $g$ into the fully connected network to obtain the final classification label through softmax function:

$$label = argmax(softmax(tanh(gW_g))) \tag{19}$$

## 5.6 Domain Center

Since there are a large number of DGA and benign domain names, when a new domain name is input, it is necessary to determine whether the domain name is a DGA domain by calculating the distances between the feature vector of the domain name and the feature vectors of all domain names in the training set. Therefore, in order to reduce the DGA detection time on the validation set, the Domain Center is constructed. The Domain Center stores the mean vectors of the hidden vectors obtained by the deep learning model for all samples of different categories. When a new domain name is input, it only needs to judge the Euler distances [18]

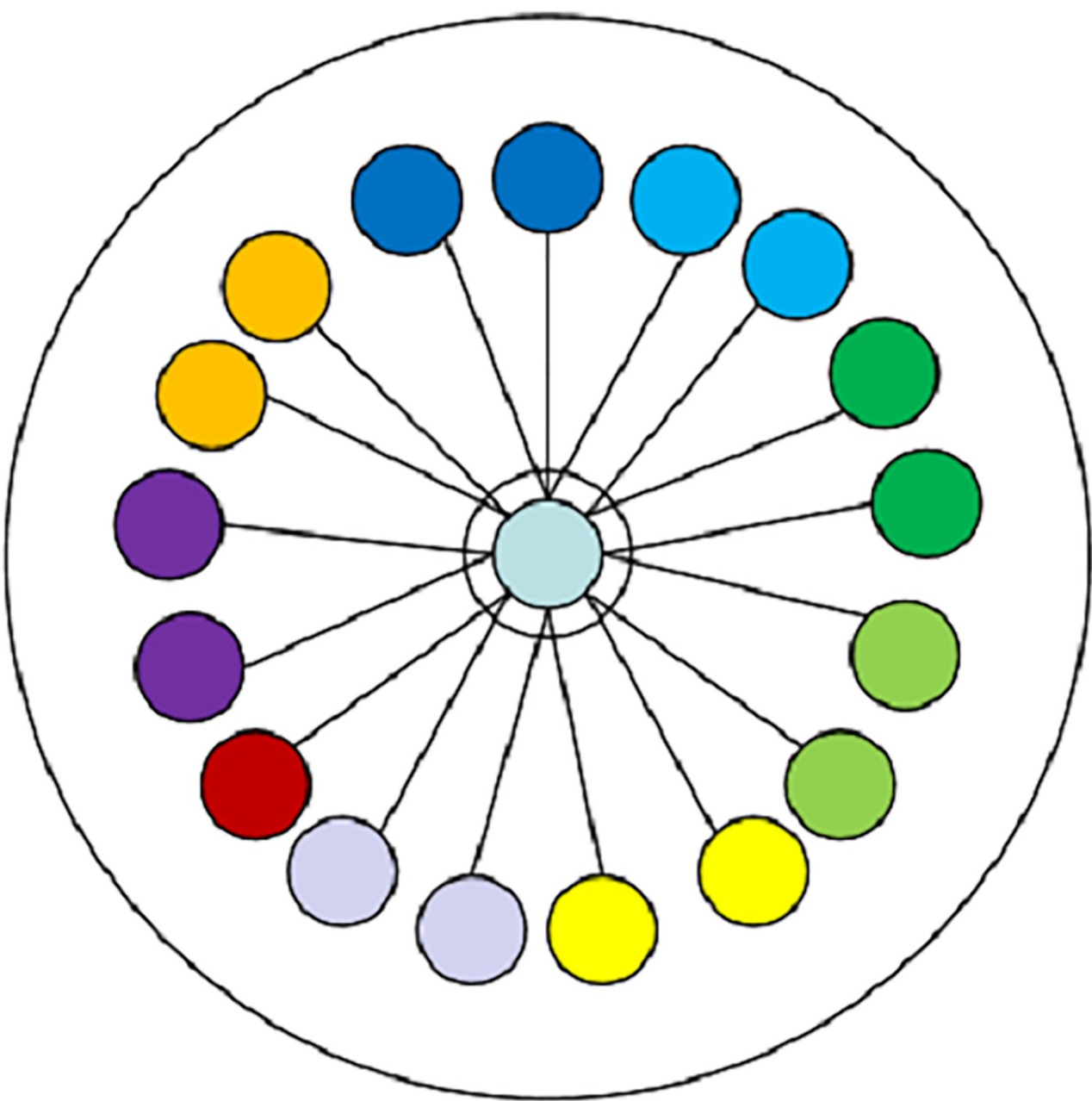

**Fig 7. The process of original DGA detection method.**

between the hidden vector of the domain name and the mean vectors stored in the Domain Center, thereby realizing fast domain name classification. The original and improved DGA detection methods are shown in Figs 7 and 8, respectively.

The processes of the original and Domain Center methods are as follows. First, all samples in the training set $X = \{X_1, X_2, \cdots, X_l\}$ are stored in the Domain Center, $l$ denotes the number of samples, $X_i = \{x_{i1}, x_{i2}, \cdots, x_{in}\}$, $n$ is the length of the feature dimension. Then, suppose that the feature vector of the domain name to be detected is $Y = \{y_1, y_2, \cdots, y_n\}$. The process of

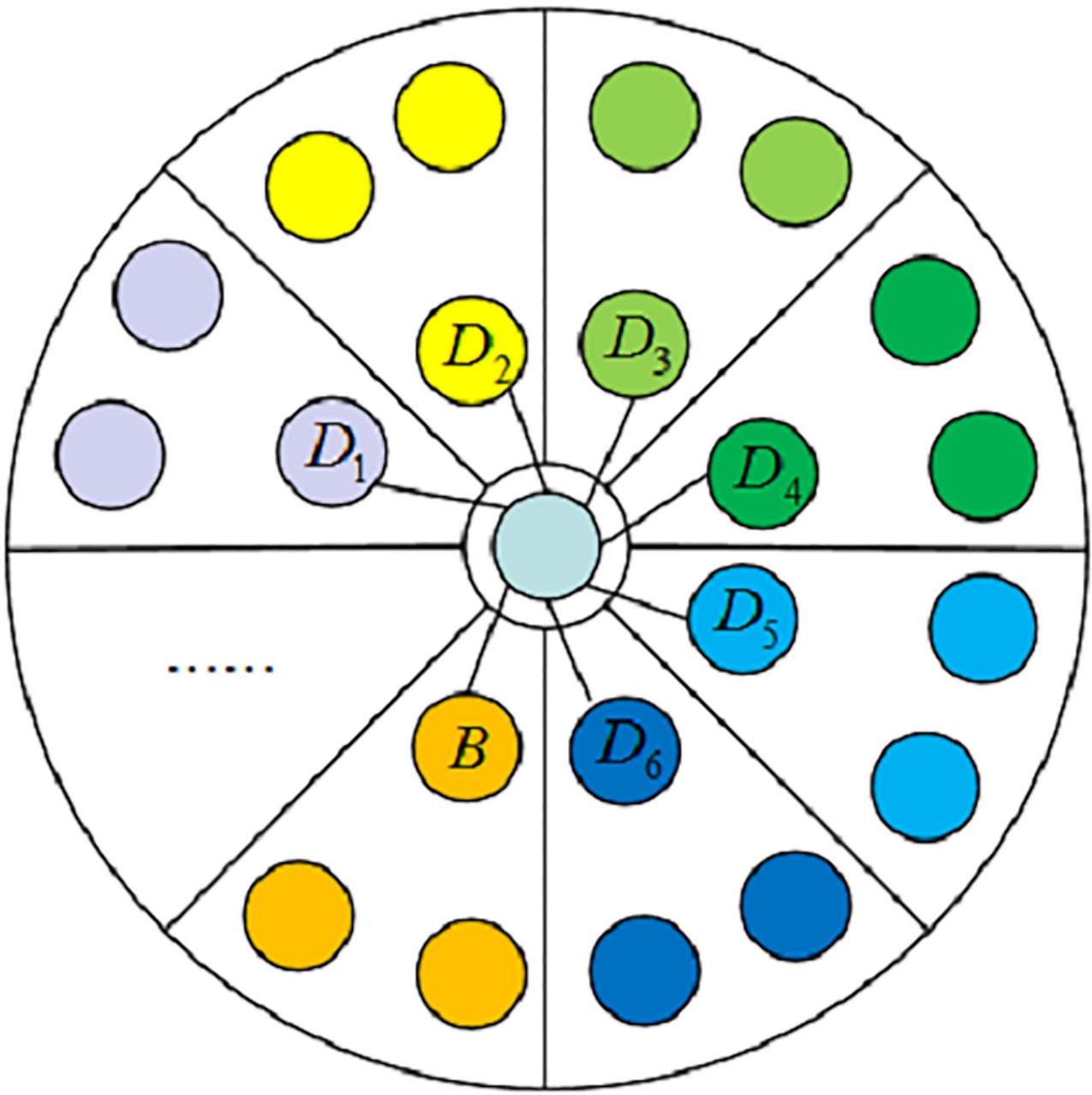

**Fig 8. The process of Domain Center.**

original DGA detection method is as follows.

$$Test_i = \sqrt{(x_{i1} - y_1)^2 + (x_{i2} - y_2)^2 + ... + (x_{in} - y_n)^2}$$
$$y_{label} = x_{label} min(Test_i) \tag{20}$$

It can be seen that the time complexity of traditional DGA detection is $o(l \times n)$. The Domain Center is proposed to reduce detection time and the processes of Domain Center are as follows.

First, the benign mean vector $B$ and DGA mean vectors $D_1$-$D_c$ stored in the Domain Center are calculated, $c$ denotes the number of categories of DGA domains:

$$B = \{b_1, b_2, ..., b_n\}$$

$$b_i = \frac{1}{l_b}\sum_{i=1}^{l_b}x_i, \ X_{label} = 0$$

$$D_1 = \{d_{11}, d_{12}, \cdots, d_{1n}\}$$

$$d_{1i} = \frac{1}{l_1}\sum_{i=1}^{l_1}x_{1i}, \ X_{label} = 1$$

$$D_2 = \{d_{21}, d_{22}, \cdots, d_{2n}\}$$

$$d_{2i} = \frac{1}{l_2}\sum_{i=1}^{l_2}x_{2i}, \ X_{label} = 2$$

$$......$$

$$D_a = \{d_{a1}, d_{a2}, \cdots, d_{an}\}$$

$$d_{ai} = \frac{1}{l_c}\sum_{i=1}^{l_c}x_{ci}, \ X_{label} = c$$

(21)

Where $l_b$ denotes the number of benign domains. $X_{label} = 0$ denotes the benign domains. $l_i$ denotes the number of DGA domains labeled $i$, $i = 1, 2, \cdots, c$ indicates the class of the DGA. The improved similarity is calculated as follows:

$$Test_B = \sqrt{(b_1 - y_1)^2 + (b_2 - y_2)^2 + ... + (b_n - y_n)^2}$$

$$Test_{T_1} = \sqrt{(d_{11} - y_1)^2 + (d_{12} - y_2)^2 + ... + (d_{1n} - y_n)^2}$$

$$Test_{T_2} = \sqrt{(d_{21} - y_1)^2 + (d_{22} - y_2)^2 + ... + (d_{2n} - y_n)^2}$$

$$......$$

$$Test_{T_c} = \sqrt{(d_{c1} - y_1)^2 + (d_{c2} - y_2)^2 + ... + (d_{cn} - y_n)^2}$$

$$y_{label} = x_{label}min(Test_B, Test_{T_1}, Test_{T_2}, ..., Test_{T_c})$$

(22)

It can be seen that the time complexity of the Domain Center is $o(c+ 1)$, which will greatly reduce the time of domain name classification.

## 6 Experiment

### 6.1 Dataset

DGA domains: 58,000 domain names are selected from 360netlab, which exclude 100,000 domain names used to build DNBSFS. The dataset contains all 58 DGA families such as tordwm, dircrypt and fobber, and each DGA family contains 1,000 domain names.

Benign domain names: 58,000 domain names are selected from Alex top 1 million as benign domain names, which exclude 100,000 domain names used to build DNWSFS.

### 6.2 Baseline methods

WSDL [7]: This model proposes a set of heuristic algorithm, which automatically marks the domain names monitored in the real traffic through the weakly supervised deep learning algorithm.

HDNN [8]: This model adopts an improved parallel CNN architecture with multi-scale convolution kernels to extract multi-scale local features from domain names. The framework also includes a BiLSTM architecture based on self-attention, which can extract bi-directional global features with Attention mechanism from domain names.

DNSML [36]: This model selects five characteristics of domain names and uses Random Forest to detect DGA domains.

DBD [37]: This model obtains the implicitly extracted statistical features and classifies domain names through deep learning architecture.

CNN-BiLSTM [29]: This model further maximizes the detection performance by using the CNN + BiLSTM integrated model, and allows the model to learn local and global informations in the domain sequence at the same time.

DGA-RNN [31]: This model constructs the feature set of domain names and uses simple RNN to detect DGA domains.

N-CBDC [32]: This model combines N-gram and deep RNN for DGA classification.

ATT-CNN-BiLSTM [33]: This model first uses CNN and BiLSTM to extract the hidden feature of the feature vector, and then uses the Attention network to assign different weights to the hidden feature.

LA-BM07 [28]: This model constructs two deep learning models by combining the LSTM network and Attention layer. The model can judge whether a domain name is benign or malicious, and can identify the DGA families of malicious domain names.

HAGD [30]: This model constructs three feature extraction methods adapted to the length of the domain name. The model adopts different detection methods for different lengths of domain names.

The above baseline methods are reproduced according to the references, and the optimal results of each baseline method are obtained through parameter tuning.

## 6.3 Experimental environment

Two layers of BiLSTM and three layers of CNN are included in this paper. The neurons of the two BiLSTM networks are 128 and 256, respectively, and 0.3 dropout is used at the end of the second layer. Each layer of CNN consists of a convolutional layer and a max pooling layer, and a droupout of 0.3 is used after each pooling layer. The sizes of the three-layer convolution kernel are all $3 \times 3$, and the numbers are 128, 64, 64 respectively. The sizes of the pooling template are all 3. The epoch is 150, the batch size is 32, and the learning rate is 0.00001.

## 6.4 Evaluation criteria

Accuracy, Precision, Recall and F1 are used as evaluation criteria, which are calculated as follows.

$$Accuracy = \frac{TP + TN}{TP + FP + TN + FN}$$

$$Precision = \frac{TP}{TP + FP}$$

(23)

$$Recall = \frac{TP}{TP + FN}$$

$$F1 = \frac{2 * Precision * Recall}{Precision + Recall}$$

Where $P$ and $N$ respectively represent the number of DGA and benign domains, $TP$ and $TN$ respectively represent the number of correctly predicted DGA and benign domains, and $FP$ and $FN$ respectively represent the number of incorrectly predicted DGA and benign domains.

## 6.5 Experimental results

FEDCC and baseline methods are applied to DGA and benign domain name datasets respectively, and the DGA detection results are shown in Table 5. It can be seen from Table 5 that FEDCC obtains the best classification Accuracy, Precision, Recall and F1, which are 0.9713, 0.9627, 0.9765 and 0.9696, respectively. In the baseline methods, the Accuracy, Precision, Recall and F1 of HAGD, LA-BM07, ATT-CNN-BiLSTM and HDNN are greater than 0.9, the Accuracy, Precision, Recall and F1 of CNN-BiLSTM, DGA-RNN and N-CBDC are between 0.8–0.9, and these of the remaining methods are less than 0.8. Among them, DNSML obtains the worst Accuracy, Precision, Recall and F1, which are 0.2394, 0.2368, 0.2893 and 0.2636 lower than FEDCC respectively. HAGD obtains the best Accuracy, Precision, Recall and F1, which are 0.0191, 0.0166, 0.0264 and 0.0215 lower than FEDCC respectively. In terms of classification time, FEDCC obtains the optimal classification time, which is 1.3s. In the baseline methods, ATT-CNN-BiLSTM obtains the longest classification time, which is 113.2s, 87 times that of FEDCC. DNSML obtains the shortest classification time, which is 15.7s, 12 times that of FEDCC.

Although most baseline methods use deep learning models as classifiers, and some baseline methods use a variety of deep learning models to build complex neural networks, FEDCC still has greatly improved the DGA detection results. We analyze the reasons. WSDL, DNSML and DBD use simple deep learning model with fewer features, which obtain the worst Accuracy, Precision, Recall and F1. Although CNN-BiLSTM, DGA-RNN and N-CBDC jointly use multiple deep learning models, since the features extracted by them are still limited, the Accuracy, Precision, Recall and F1 of them are not that ideal. HDNN, ATT-CNN-BiLSTM, LA-BM07, and HAGD use more complex deep learning models and features that are not rich enough. Although their Accuracy, Precision, Recall and F1 are greater than 0.9, they are still lower than that of FEDCC. In addition to the domain name features, FEDCC not only obtains the Whois features, but also obtains the N-gram features by constructing the DNWSFS and DNBSFS. Through rich feature acquisition and complex deep learning model design, FEDCC greatly improves the Accuracy, Precision, Recall and F1. In addition, the time complexity is reduced

**Table 5. DGA detection results of comparison models.**

| Methods | Accuracy | Precision | Recall | F1 | time(s) |
|---|---|---|---|---|---|
| WSDL | 0.7981 | 0.7851 | 0.7993 | 0.7926 | 43.8 |
| HDNN | 0.9269 | 0.9177 | 0.9241 | 0.9209 | 69.7 |
| DNSML | 0.7319 | 0.7259 | 0.6872 | 0.7060 | 15.7 |
| DBD | 0.7699 | 0.7407 | 0.713 | 0.7266 | 48.6 |
| CNN-BiLSTM | 0.8497 | 0.8514 | 0.8452 | 0.8483 | 76.9 |
| DGA-RNN | 0.8167 | 0.8183 | 0.8124 | 0.8153 | 38.7 |
| N-CBDC | 0.8369 | 0.8277 | 0.8398 | 0.8337 | 49.2 |
| ATT-CNN-BiLSTM | 0.9408 | 0.9332 | 0.9288 | 0.9310 | 113.2 |
| LA-BM07 | 0.9328 | 0.9455 | 0.9317 | 0.9385 | 86.6 |
| HAGD | 0.9522 | 0.9461 | 0.9501 | 0.9481 | 102.7 |
| FEDCC | 0.9713 | 0.9627 | 0.9765 | 0.9696 | 1.3 |

from $O(l \times n)$ to $O(c + 1)$ by constructing the Domain Center, which greatly reduces the classification time.

## 6.6 Model analysis

**6.6.1 Receiver Operating Characteristic (ROC) curves.** The DGA detection dataset constructed in Section 6.1 belongs to the binary classification balance dataset. Therefore, in order to better interpret the DGA detection results of FEDCC and baseline methods, the ROC curves of the detection results of each comparison models are drawn in Fig 9, and the Area Under the Curve (AUC) values of each model are calculated. It is easy to see from the ROC curves and AUC values that FEDCC is significantly better than all baseline methods.

**6.6.2 Classification results of different DGA families.** In Section 6.1, the domain names of 58 DGA families are uniformly assigned to the DGA domain dataset, and the domain name naming methods of different DGA families may be quite different. For example, the domain names of the Banjori, Chinad, Conficker families are constructed based on random characters, while the domain names of the Bigviktor, Matsnu families are constructed based on dictionaries. To further verify the classification effect of FEDCC on different DGA families, the following experiments are conducted. 1000 benign domain names and the domain names of 58 DGA families in Section 6.1 are selected to form the new domain name datasets. FEDCC and baseline methods are applied to these datasets to obtain the detection results of 58 DGA

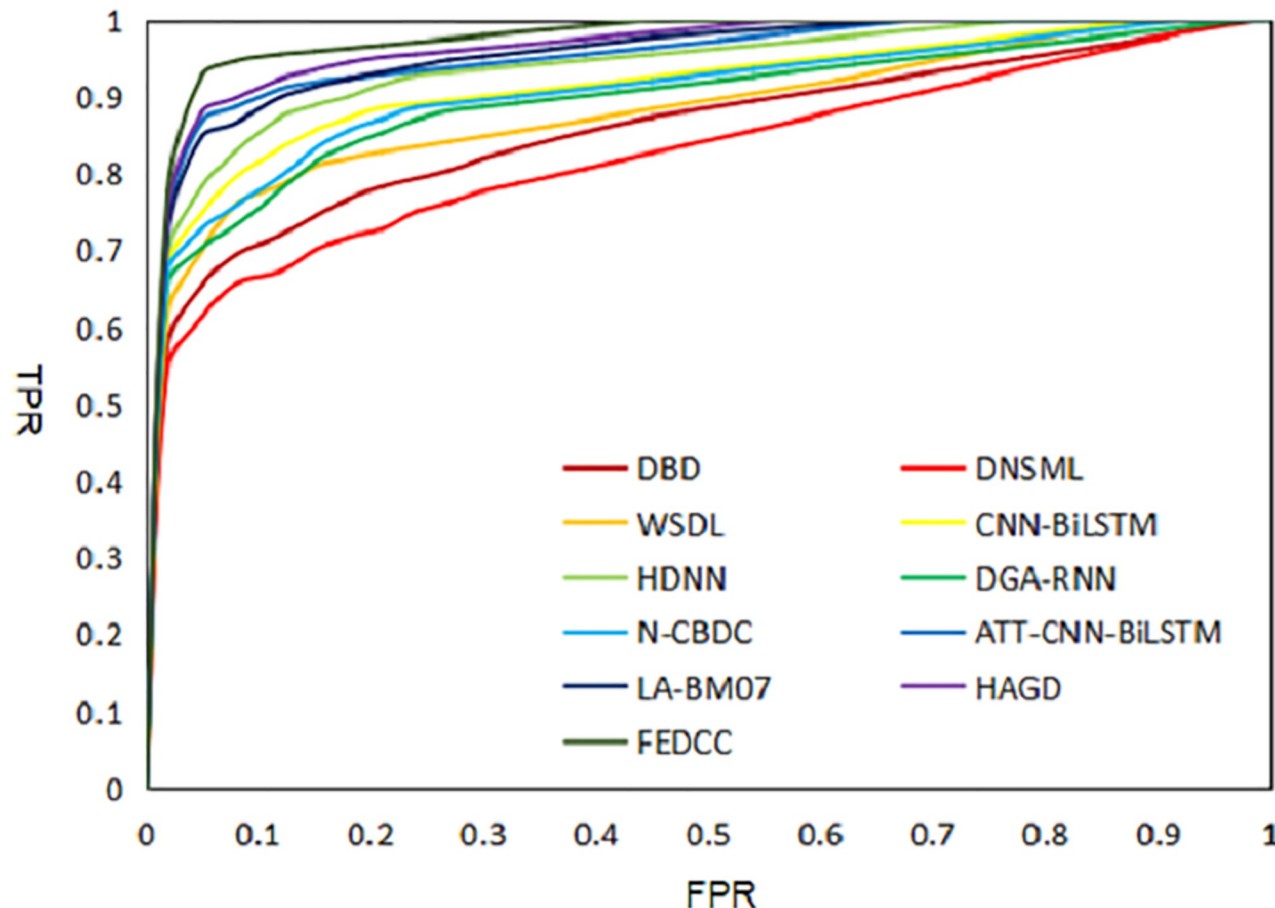

**Fig 9. ROC curves of DGA detection results of comparison models.**

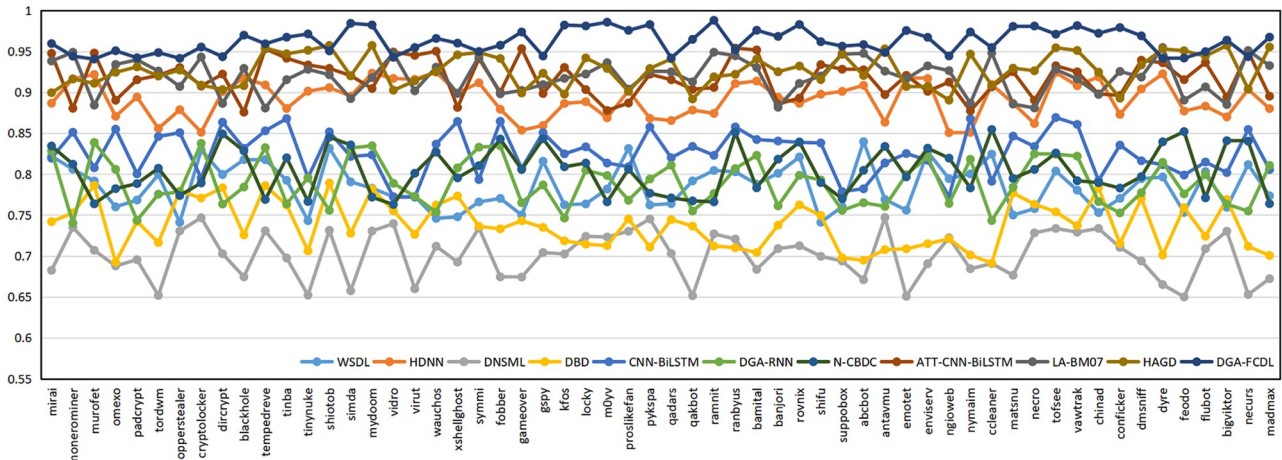

**Fig 10. The classification Accuracy of the comparison models for 58 DGA families.**

families, and the classification Accuracy, Precision, Recall and F1 are shown in Figs 10–13, respectively. From Figs 10–13, it can be seen that the classification results of FEDCC in most DGA families are far better than the baseline methods. Therefore, FEDCC can not only accurately detect the DGA domains, but also accurately judge which DGA family the DGA domain belongs to.

**6.6.3 Importance analysis of each component.** A feature set with rich features, including the domain name features, Whois features and the N-gram features is constructed in this paper. In addition, a deep learning model based on BiLSTM, Attention, skip connect and CNN is built. Additionally, the Domain Center is built for fast DGA detection. Through the previous experiments, it can be seen that the model constructed in this paper greatly promotes the DGA detection results. The following experiments are conducted to verify the influence of each component. The relative improvement ratio Δ, the classification accuracy and the classification time are used as the evaluation metrics:

$$\Delta = (ACC_{FEDCC} - ACC) \div ACC$$

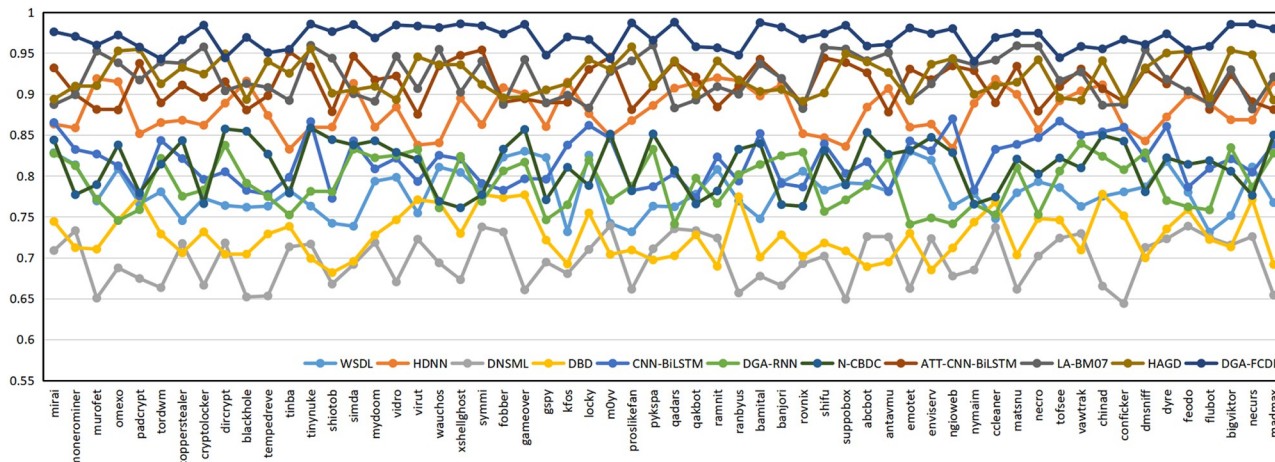

**Fig 11. The classification Precision of the comparison models for 58 DGA families.**

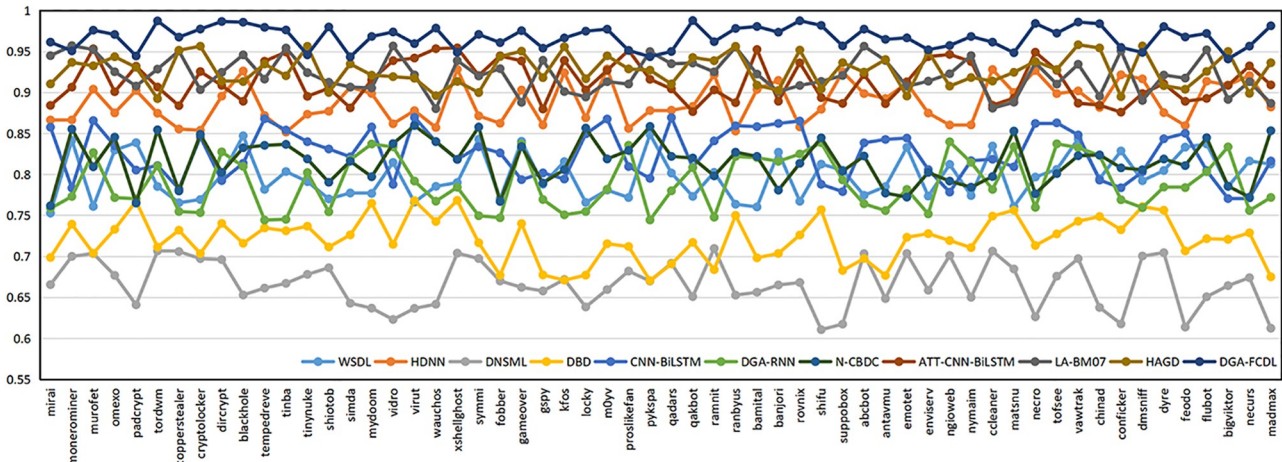

**Fig 12. The classification Recall of the comparison models for 58 DGA families.**

Where $ACC_{FEDCC}$ represents the DGA detection accuracy of FEDCC, and $ACC$ represents the DGA detection Accuracy after changing each component. The results are shown in Table 6. It can be seen from Table 6 that when any component of FEDCC is removed, the detection Accuracy is reduced, and the removal of a feature set reduces the Accuracy of the model to a greater extent than the removal of the Domain Center or a component of the deep learning model. By analyzing the three feature sets, it can be seen that Whois features has the greatest impact on the model, followed by the N-gram features and the domain name features. By analyzing the four components of the deep learning model, it can be seen that BiLSTM has the greatest impact on the model, followed by the Attention, skip connect and CNN.

Although the Domain Center is less important in classification accuracy than the feature set, when it is removed, classification time increases substantially. Therefore, the Domain Center can greatly reduce classification time while improving classification accuracy.

**6.6.4 Importance analysis of different features.** Domain name features, Whois features and N-gram features are obtained to construct a feature vector with a length of 35. Through the above experiments, we know that the construction of the feature vector plays a great role in

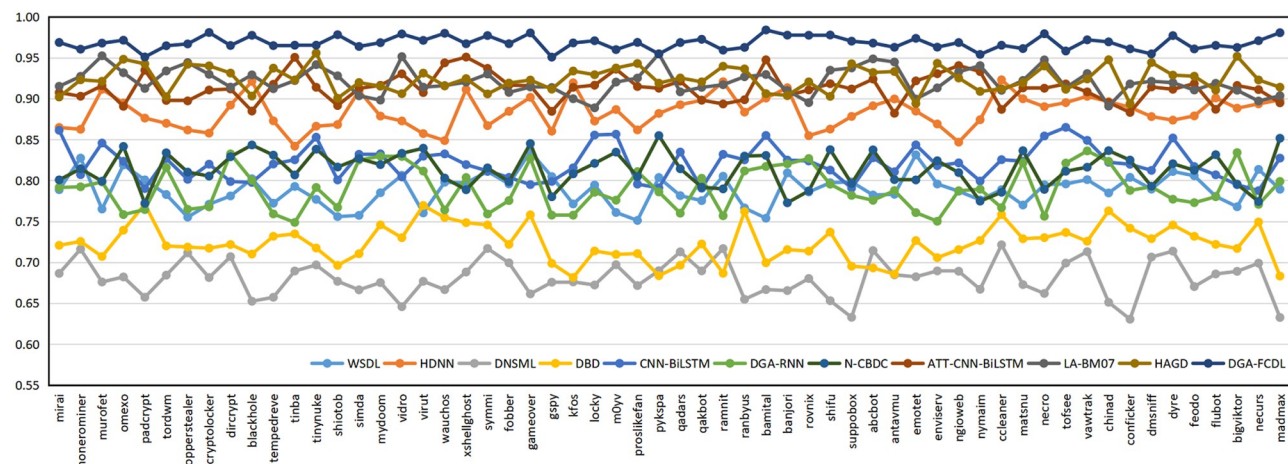

**Fig 13. The classification F1 of the comparison models for 58 DGA families.**

**Table 6. DGA detection results of different variant models.**

| Variant Model | Accuracy | Δ | Time(s) |
|---|---|---|---|
| Remove domain name features | 0.9478 | 0.0248 | 1.3283 |
| Remove Whois features | 0.9401 | 0.0332 | 1.314 |
| Remove N-gram features | 0.9436 | 0.0294 | 1.3217 |
| Remove BiLSTM | 0.9571 | 0.0148 | 0.8746 |
| Remove Attention | 0.9593 | 0.0125 | 1.1033 |
| Remove CNN | 0.9648 | 0.0067 | 0.9027 |
| Remove skip connect | 0.9625 | 0.0091 | 1.285 |
| Remove Domain Center | 0.9486 | 0.0239 | 94.68 |

promoting the DGA detection results. In order to further study the influence of different features on the detection results, the importance of different features is analyzed through the following experimentS. We refer to the official document of LightGBM to calculate the importanceS of 35 features, and the calculation formulas are as follows:

$$feature-importance_F = \sum\nolimits_{trees,leafs_F} (v_1 - avr)^2 c_1 + (v_2 - avr)^2 c_2 \quad avr = \frac{v_1 c_1 + v_2 c_2}{c_1 + c_2}$$

Where $c_1$ and $c_2$ are the number of objects in each leaf respectively, $v_1$ and $v_2$ are the formula values in the left and right leaves respectively. The importance results of 35 features are shown in Fig 14. It can be seen from Fig 14 that most of the Whois features are more important than the domain name features and the N-gram features, which further verifies the experimental results in Section 6.6.3. Analyzing the reason, Whois contains entity information such as domain name registrar and server, and the extraction of these information greatly improves the DGA detection results.

**6.6.5 Analysis of the Domain Center.** The Domain Center is proposed to quickly detect whether the newly entered domain name is a DGA domain through Euler distance [18]. Since there are many methods to calculate the vector similarity, the following experiments are carried out to explain why Euler distance is used in this paper. Besides Euler distance, Manhattan distance [38], cosine similarity [39], Minkowski distance [40] and Chebyshev distance [41] are also selected, and the experimental results of different similarity methods are shown in Table 7. It can be seen that although Manhattan distance uses less time and Chebyshev distance achieves the highest Precision, Euler distance achieves the overall optimal results. Therefore, Euler distance is selected to quickly classify DGA domains.

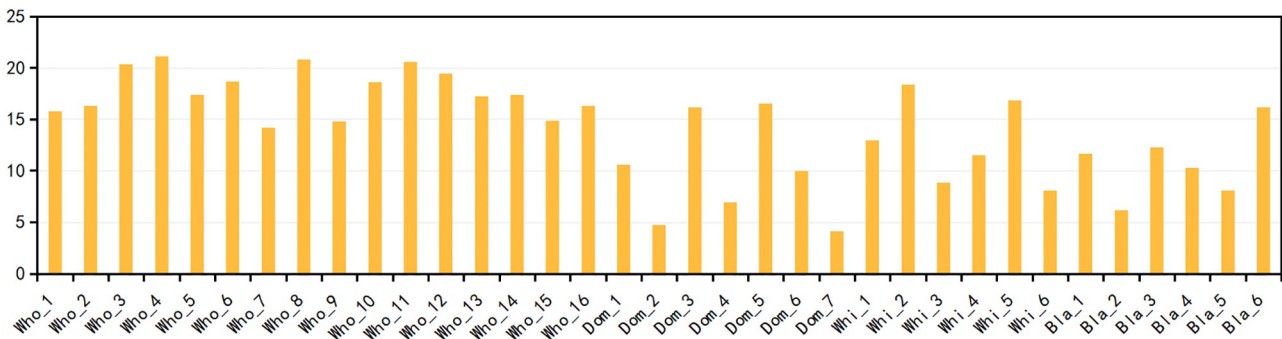

**Fig 14. Importance results of different features.**

**Table 7. DGA detection results of different similarity methods.**

| Distance | Accuracy | Precision | Recall | F1 | Time (s) |
|---|---|---|---|---|---|
| Manhattan distance | 0.9587 | 0.9511 | 0.9426 | 0.9468 | 1.3027 |
| cosine similarity | 0.9251 | 0.9077 | 0.918 | 0.9128 | 2.6835 |
| Minkowski distance | 0.9611 | 0.9386 | 0.9542 | 0.9463 | 2.257 |
| Chebyshev distance | 0.9521 | 0.9633 | 0.9685 | 0.9659 | 1.4758 |
| Euler distance | 0.9713 | 0.9627 | 0.9765 | 0.9696 | 1.3126 |

**Table 8. DGA detection results of different number of samples.**

| Number of samples per category | Accuracy | Precision | Recall | F1 |
|---|---|---|---|---|
| 400 | 0.9432 | 0.9367 | 0.9415 | 0.9391 |
| 600 | 0.9589 | 0.9527 | 0.9603 | 0.9565 |
| 800 | 0.9658 | 0.9587 | 0.972 | 0.9653 |
| 1000 | 0.9713 | 0.9627 | 0.9765 | 0.9696 |
| 1200 | 0.9738 | 0.9647 | 0.9785 | 0.9716 |
| 1500 | 0.9758 | 0.9659 | 0.9793 | 0.9726 |
| 2000 | 0.9771 | 0.9668 | 0.9802 | 0.9735 |
| 3000 | 0.9773 | 0.9669 | 0.9804 | 0.9736 |

The mean vectors of the hidden vectors of the datasets in Section 6.1 are stored in the Domain Center. The difference in the number of samples in the training set will cause the difference in the mean vectors stored in the Domain Center, which in turn affects the DGA detection results. The following experiments verify the relationship between the DGA detection results and the amount of samples in the training set, and the experimental results are shown in Table 8. It can be seen that as the number of samples in each category increases, the DGA detection results gradually improve. When the number of samples in each category exceeds 1,000, the increase of DGA detection results with the increase of the number of samples becomes smaller and smaller, but the training time is proportional to the amount of samples. Therefore, the number of samples for each category is set to 1000 in this paper.

## 7 Conclusion

Due to the limited features that the traditional DGA detection model can extract, the DGA detection results are not that ideal. In order to solve the above problem, rich feature set, including the domain name features, the Whois features and the N-gram features, and a deep learning model based on BiLSTM, Attention and CNN are constructed for DGA detection. In addition, the Domain Center is built to reduce the DGA detection time. Multiple comparative experiment results prove that the proposed model not only gets the best Accuracy, Precision, Recall and F1, but also greatly reduces the classification time of newly entered domain names.

However, the model built in this paper is based on passively acquired data and cannot actively detect DGA domains. In order to better realize the task of DGA detection and governance, our main task in the future is to develop a model to actively detect DGA, so as to actively prevent the infringement of DGA domains.

## Supporting information

**S1 Data.**
(RAR)

## Author Contributions

**Conceptualization:** Xinjie Sun.

**Data curation:** Xinjie Sun.

**Formal analysis:** Xinjie Sun.

**Funding acquisition:** Xinjie Sun.

**Investigation:** Xinjie Sun, Zhifang Liu.

**Methodology:** Xinjie Sun.

**Project administration:** Xinjie Sun.

**Resources:** Xinjie Sun.

**Software:** Xinjie Sun, Zhifang Liu.

**Supervision:** Xinjie Sun.

**Validation:** Xinjie Sun.

**Visualization:** Xinjie Sun, Zhifang Liu.

**Writing – original draft:** Xinjie Sun, Zhifang Liu.

**Writing – review & editing:** Xinjie Sun.

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
