## [Decision Letter · Decision Letter 0]

26 Oct 2022

PONE-D-22-26290DGA Detection with Feature Extraction and Domain Center ConstructionPLOS ONE

Dear Dr. Sun,

Thank you for submitting your manuscript to PLOS ONE. After careful consideration, we feel that it has merit but does not fully meet PLOS ONE’s publication criteria as it currently stands. Therefore, we invite you to submit a revised version of the manuscript that addresses the points raised during the review process.

We look forward to receiving your revised manuscript.

Kind regards,

Kamran Siddique

Academic Editor

PLOS ONE

Journal Requirements:

2. In your Methods section, please include additional information about your dataset and ensure that you have included a statement specifying whether the collection and analysis method complied with the terms and conditions for the source of the data

"Natural Science Foundation of Shandong Province [Grant No. ZR2020KF009]. National 

Science Research Project of Department of Education in Guizhou Province [Grant nos. 

KY[2020]112]"

"Natural Science Foundation of Shandong Province [Grant No. ZR2020KF009]. National Science Research Project of Department of Education in Guizhou Province [Grant nos. KY[2020]112]"

Additional Editor Comments:

The paper is interesting and written well. Some minor issues are indicated in the reviews. The authors are encourage to address them for further improvements.

Reviewers' comments:

Reviewer's Responses to Questions

**Comments to the Author**

1. Is the manuscript technically sound, and do the data support the conclusions?

Reviewer #1: Yes

Reviewer #2: Yes

Reviewer #3: Partly

2. Has the statistical analysis been performed appropriately and rigorously? 

Reviewer #1: Yes

Reviewer #2: Yes

Reviewer #3: Yes

3. Have the authors made all data underlying the findings in their manuscript fully available?

Reviewer #1: Yes

Reviewer #2: Yes

Reviewer #3: Yes

4. Is the manuscript presented in an intelligible fashion and written in standard English?

Reviewer #1: Yes

Reviewer #2: Yes

Reviewer #3: Yes

5. Review Comments to the Author

Reviewer #1: In this study, a BiLSTM-based model is proposed using Whois, N-gram and domain name features. The architecture of the model has been dealt with effectively. The model proposed by the authors yielded a high accuracy rate. At the same time, the classification time is very short compared to other methods.

Reviewer #2: General Evaluation: The paper deals with an interesting topic and it seems it is not enough mature yet.

Using an abbreviation (DGA) in the title should be avoided since it makes it difficult to understand.

Overall, this is a clear, concise, and well-written manuscript. The introduction is to the point and based on theory. Sufficient information on the results of the previous study is presented.

Reviewer #3: 1) In the abstract section, after the problem definition, the aim of study was not emphasized.

The sentence starting with "Therefore..." may be continued as " In order to overcome this problem, in this study..." or similar.

Similarly, in the conclusion section at line 425, the aim and results of the study was not emphasized. Please check it.

2) The authors should add a paper outline at the end of the introduction section.

3) The authors should fix the figure numbers at line 358. (??-??)

4) The performance of the study should be compared with similar studies in the literature (For example reference [37]). The pros and cons should be examined.

5) The authors should add more references for the literature review section.

6. PLOS authors have the option to publish the peer review history of their article (what does this mean?). If published, this will include your full peer review and any attached files.

Reviewer #1: **Yes: **Salih Taha Alperen Ozcelik

Reviewer #2: No

Reviewer #3: No

---

## [Author Response · Author response to Decision Letter 0]

6 Dec 2022

I. Reviewer1

1.In this study, a BiLSTM-based model is proposed using Whois, N-gram and domain name features. The architecture of the model has been dealt with effectively. The model proposed by the authors yielded a high accuracy rate. At the same time, the classification time is very short compared to other methods.

R. Thank you for your approval.

Ⅱ. Reviewer2

1.General Evaluation: The paper deals with an interesting topic and it seems it is not enough mature yet.

Using an abbreviation (DGA) in the title should be avoided since it makes it difficult to understand.

Overall, this is a clear, concise, and well-written manuscript. The introduction is to the point and based on theory. Sufficient information on the results of the previous study is presented.

R. Thank you for your approval. We changed ‘DGA’ in the title to ‘Domain Generation Algorithms’.

Ⅲ. Reviewer3

1.In the abstract section, after the problem definition, the aim of study was not emphasized.

The sentence starting with "Therefore..." may be continued as " In order to overcome this problem, in this study..." or similar.

Similarly, in the conclusion section at line 425, the aim and results of the study was not emphasized. Please check it.

R. We changed ‘Therefore’ in the abstract to ‘In order to overcome this problem’.

We added ‘Due to the limited features that the traditional DGA detection model can extract, the DGA detection results are not that ideal. In order to solve the above problem’ at line 425.

2.The authors should add a paper outline at the end of the introduction section.

R. We added ‘The remainder of this work is organized as follows. Section 2 introduces the latest research results of DGA detection; Section 3 introduces the background of BILSTM, Attention mechanism and CNN; Section 4 introduces the construction method of the feature set; Section 5 introduces the structure of the deep learning model constructed in this paper; Section 6 introduces the data set selected in this paper and the experimental results; Section 7 provides a final conclusion.’ at the end of the introduction section.

3. The authors should fix the figure numbers at line 358. (??-??)

R. We corrected this error.

4. The performance of the study should be compared with similar studies in the literature (For example reference [37]). The pros and cons should be examined.

R. We changed ‘As can be seen from Table 5, compared with the optimal baseline methods, FEDCC obtained the best classification results. Specifically, the classification Accuracy, Precision, Recall and F1 have been improved at least 0.0191, 0.0166, 0.0264 and 0.0215 respectively, and the classification time is only 1.3s, which is far lower than the optimal classification time of the baseline methods.’ in the first paragraph of Section 6.5 to ‘It can be seen from Table 5 that FEDCC obtains the best classification Accuracy, Precision, Recall and F1, which are 0.9713, 0.9627, 0.9765 and 0.9696, respectively. In the baseline methods, the Accuracy, Precision, Recall and F1 of HAGD, LA-BM07, ATT-CNN-BiLSTM and HDNN are greater than 0.9, the Accuracy, Precision, Recall and F1 of CNN-BiLSTM, DGA-RNN and N-CBDC are between 0.8-0.9, and these of the remaining methods are less than 0.8. Among them, DNSML obtains the worst Accuracy, Precision, Recall and F1, which are 0.2394, 0.2368, 0.2893 and 0.2636 lower than FEDCC respectively. HAGD obtains the best Accuracy, Precision, Recall and F1, which are 0.0191, 0.0166, 0.0264 and 0.0215 lower than FEDCC respectively. In terms of classification time, FEDCC obtains the optimal classification time, that is, 1.3s. In the baseline methods, ATT-CNN-BiLSTM obtains the longest classification time, which is 113.2s, 87 times that of FEDCC. DNSML obtains the shortest classification time, which is 15.7s, 12 times that of FEDCC.’

We changed ‘e analyze the reasons. The features obtained in baseline methods are very limited, and almost all of them only obtain the domain name features. Due to the randomness of the setting of the domain name, only selecting the features of the domain name limits the DGA detection results to a great extent. ’ in the second paragraph of Section 6.5 to ‘We analyze the reasons. WSDL, DNSML and DBD use simple deep learning model with fewer features, which obtain the worst Accuracy, Precision, Recall and F1. Although CNN-BiLSTM, DGA-RNN and N-CBDC jointly use multiple deep learning models, since the features extracted by them are still limited, the Accuracy, Precision, Recall and F1 of them are not that ideal. HDNN, ATT-CNN-BiLSTM, LA-BM07, and HAGD use more complex deep learning models and features that are not rich enough. Although their Accuracy, Precision, Recall and F1 are greater than 0.9, they are still lower than that of FEDCC.’

5. The authors should add more references for the literature review section.

R. We added ‘Chin et al. [25] proposed a machine learning framework for identifying and clustering domain names to circumvent threats from the DGAs. Li et al. [26] proposed a machine learning framework (a two-level model and a prediction model) for identifying and detecting DGA domains to alleviate the threat of them. Baruch et al. [27] surveyed different machine learning methods for detecting DGAs by analyzing only the alphanumeric characteristics of the domain names in the network.’ at the end of the second paragraph of Section 2.

We added ‘Lison et al. [31] demonstrated that a deep learning approach based on RNN was able to detect domain names generated by DGAs with high precision. Xu et al. [32] combined n-gram and a deep CNN to propose a novel n-gram combined character based domain classification (N-CBDC) model. Experiments on real-world data showed that N-CBDC could effectively detect DGAs. Ren et al. [33] proposed a deep learning framework for identifying and detecting DGA domains.’ at the end of the third paragraph of Section 2.

---

## [Decision Letter · Decision Letter 1]

19 Dec 2022

Domain Generation Algorithms Detection with Feature Extraction and Domain Center Construction

PONE-D-22-26290R1

Dear Dr. Sun,

We’re pleased to inform you that your manuscript has been judged scientifically suitable for publication and will be formally accepted for publication once it meets all outstanding technical requirements.

Kind regards,

Kamran Siddique

Academic Editor

PLOS ONE

Additional Editor Comments (optional):

Reviewers' comments:

Reviewer's Responses to Questions

**Comments to the Author**

1. If the authors have adequately addressed your comments raised in a previous round of review and you feel that this manuscript is now acceptable for publication, you may indicate that here to bypass the “Comments to the Author” section, enter your conflict of interest statement in the “Confidential to Editor” section, and submit your "Accept" recommendation.

Reviewer #1: All comments have been addressed

Reviewer #2: All comments have been addressed

Reviewer #3: All comments have been addressed

2. Is the manuscript technically sound, and do the data support the conclusions?

Reviewer #1: Yes

Reviewer #2: Yes

Reviewer #3: Yes

3. Has the statistical analysis been performed appropriately and rigorously? 

Reviewer #1: Yes

Reviewer #2: Yes

Reviewer #3: Yes

4. Have the authors made all data underlying the findings in their manuscript fully available?

Reviewer #1: Yes

Reviewer #2: Yes

Reviewer #3: Yes

5. Is the manuscript presented in an intelligible fashion and written in standard English?

Reviewer #1: Yes

Reviewer #2: Yes

Reviewer #3: Yes

6. Review Comments to the Author

Reviewer #1: The authors' consideration of reviewer 2 and reviewer 3's recommendations made the study more meaningful.

Reviewer #2: Overall, this is a clear, concise, and well-written manuscript.

The introduction is relevant and theory based.

Sufficient information about the previous study findings is presented.

Reviewer #3: The authors have made the necessary adjustments and corrected the deficiencies by adhering to the warnings and suggestions.

7. PLOS authors have the option to publish the peer review history of their article (what does this mean?). If published, this will include your full peer review and any attached files.

Reviewer #1: **Yes: **SALİH TAHA ALPEREN ÖZÇELİK

Reviewer #2: No

Reviewer #3: No

---

## [Editor Report · Acceptance letter]

2 Jan 2023

PONE-D-22-26290R1 

Domain Generation Algorithms Detection with Feature Extraction and Domain Center Construction  

Dear Dr. Sun:

I'm pleased to inform you that your manuscript has been deemed suitable for publication in PLOS ONE. Congratulations! Your manuscript is now with our production department. 

Kind regards, 

on behalf of

Dr. Kamran Siddique 

Academic Editor

PLOS ONE